# Batch Bayesian Optimization of Delayed Effects Corrections for Thompson Sampling Bandits: A Practical Tuning Algorithm for Adaptive Interventions

## Abstract

When the number of reinforcement learning episodes that can be performed to optimize a policy is severely limited, the bias-variance trade-off of bandit algorithms such as Thompson Sampling can be significantly better than that of policy gradient and value function-based methods. However, bandits have no ability to model the delayed effects of actions. In this paper, we develop a batch Bayesian optimization algorithm that learns a delayed effect correction for linear Thompson Sampling bandits. This work is motivated by the problem of tuning adaptive intervention policies where each episode corresponds to a costly and often lengthy trial involving human subjects. We show through extensive experiments in an adaptive intervention simulation environment that the proposed approach can find beneficial delayed effects correction terms under realistic constraints on the number of Bayesian optimization rounds and the batch size per round.

## 1 Introduction

There is increasing interest in using reinforcement learning methods (RL) in the healthcare setting, including in mobile health. However, the healthcare domain presents a range of challenges for existing RL methods. In particular, each reinforcement learning episode typically corresponds to a human subjects trial involving one or more participants that is both costly to conduct and may require substantial time to carry out (e.g., weeks to months). As a result, RL methods that leverage large numbers of episodes to learn better quality policies (often through simulation) are not feasible (Mnih et al., 2013).

Within the mobile health research community specifically, adaptive intervention policy learning methods have addressed severe episode count restrictions imposed by real-world research constraints by focusing on the use of bandit algorithms (Tewari & Murphy, 2017). By focusing on maximizing immediate reward, bandit algorithms have the potential to provide an improved bias-variance trade-off compared to policy gradient and state-action value function approaches (Lattimore & Szepesvari, 2017). Linear Thompson sampling bandits are a particularly promising approach due to the further application of Bayesian inference to capture model uncertainty due to data scarcity in the low episode count setting (Agrawal & Goyal, 2013).

Of course, the main drawback of bandit-like algorithms is that they have no ability to account for the delayed effects of actions (Chapelle & Li, 2011). In the adaptive behavioral intervention setting, the decision to provide specific treatment options at specific times can indeed have delayed effects on outcomes of interest through mediating processes such as habituation and engagement.

In this work, we present an algorithm with the ability to learn delayed effects corrections for linear Thompson sampling bandits under realistic constraints on both the time needed to run human subjects trials and the total number of participants required. Our proposed approach wraps a modified linear Thompson sampling bandit algorithm in a Batch Bayesian optimization method. Essentially, the Batch Bayesian optimization method optimizes the delayed effects correction terms in an outer loop based on returns provided by a Thompson sampling bandit algorithm that leverages the de-

layed effect correction term. We refer to the proposed algorithm as *BOTS* - Bayesian Optimization of Thompson Sampling.

The use of batch Bayesian optimization allows for the decomposition of the total episode budget into a number of rounds and a batch size per round. Each element of each batch is used to evaluate a different candidate delayed effect correction term. The evaluation of a delayed effect term corresponds to running a full Thompson sampling bandit adaptive intervention using one or more individuals. A batch of delayed effect correction terms can thus be assessed in parallel using a cohort of study participants. Our primary contributions are as follows:

- We propose a new algorithm for estimating delayed effects corrections for linear Thompson sampling bandits that is practically realizable in the context of adaptive health interventions.
- Our approach addresses the problem warm-starting the Thompson sampling trials by adaptively chaining the Thompson sampler prior across Batch Bayesian optimization rounds.
- We perform an extensive performance analysis of the algorithm using a just-in-time adaptive intervention simulation environment under realistic constraints on the total episode count budget, which corresponds to the number of study participants.
- We present detailed results showing how the performance of learned policies varies as individuals are allocated to rounds and batches in different ways, as well as ablation results showing the impact of ignoring delayed effects and not warm-starting Thompson sampling between rounds.

The remainder of this paper is organized as follows. We begin by presenting background and related work in Section 2. We present the proposed approach in Section 3. In Section 4 we present experiments and results. We conclude in section 5.

## 2 BACKGROUND AND RELATED WORK

In this section, we present background and related work on Thompson sampling and Bayesian optimization.

### 2.1 THOMPSON SAMPLING

In this work, we focus on the use of linear Thompson sampling (TS) contextual bandit algorithms. This approach starts with a linear model $w_a s_t + b_a$ of the reward $r_t$ at time $t$ based on the observed state $s_t$ at time $t$ and the action taken $a \in [0, ..., A]$. The primary model parameters are the weights $w_a$ and biases $b_a$ for each action $a$. The approach then places Gaussian priors $\mathcal{N}(w_a; \mu_w, \Sigma_w)$ and $\mathcal{N}(b_a; \mu_b, \Sigma_b)$ on the primary model parameters combined with a Gaussian likelihood, producing a Bayesian linear regression model for the reward. This model supports exact computation of the posterior distribution over the model parameters given observations of states and rewards (Russo et al., 2018; Agrawal & Goyal, 2013).

To select an action at each time step $t$, we first sample values for each of the model parameters from the joint parameter posterior: $w_a, b_a \sim \mathcal{N}(\mu^{t-1}, \Sigma^{t-1})$. We then select the action that produces the largest expected reward $\hat{r}_t(s_t, a) = w_a s_t + b_a$ based on the sampled parameter values.

When Thompson sampling is used in a setting that does not satisfy the contextual bandit assumptions, it, of course, has no ability to learn the long term effects of actions. One way to address this issue is to penalize the immediate reward to take into account the delayed effects of actions. Previous work has proposed the use of a significantly more complex model-based proxy reward that requires making a number of assumptions about the delayed effects process (Liao et al., 2022). By contrast, our approach iteratively learns a delayed effect correction using real returns and an outer Bayesian optimization loop.

### 2.2 BAYESIAN OPTIMIZATION

The goal of Bayesian optimization (BO) is to optimize an objective function that is expensive to evaluate. BO has many applications, for example: it has been used for learning the parameters of

complex simulators or fine-tuning the hyper-parameters of computationally expensive algorithms. In our case, we aim to optimize parameters within a Thompson sampling bandit algorithm that requires running a human subjects trial to evaluate. Bayesian optimization methods work by iteratively constructing an approximation to the true, costly optimization objective function and optimizing the approximation.

A Gaussian process (GP) regression model is typically used as the surrogate model because posterior inference in the model is exact. In this work, we use a Matérn 5/2 kernel (Snoek et al., 2012). We note that this model is stationary, since the covariance only depends on the distance between points, so it is more suitable for small dimension input space.

Candidate points are generated during Bayesian optimization using an acquisition function applied to the posterior over the known objective function. The acquisition function guides how the parameter space is explored during the Bayesian optimization process. The Probability of Improvement (PI) acquisition function selects the point $x$ with highest probability of improving the function value. This is the point $x$ that maximizes the expectation of the utility function $u(x) = 0$ if $f(x) > f^*$ and 1 otherwise. where $f^*$ is the minimum value of the function found to this point. The Upper Confidence Bound acquisition function selects the point $x$ that maximizes a function of the form $u(x) = \mu(x) - c\sigma(x)$, where $c$ is a trade-off parameter and $\mu(x)$ and $\sigma(x)$ are the posterior mean and standard deviation function of the posterior distribution over the latent objective function.

Expected Improvement (EI) is a common acquisition function based on a utility function $u(x) = \max(0, f^* - f(x))$. The EI acquisition function selects the point $x$ with the largest expected value of this utility function. In this work, we use qEI, a Monte Carlo extension of EI. qEI can tractably generate batches of multiple candidate points (Balandat et al., 2020) and thus enables the application of batch Bayesian optimization methods instead of purely sequential Bayesian optimization.

## 3 Methods

In this section we introduce our proposed algorithm for learning delayed effect corrections for Thompson Sampling bandits: BOTS. BOTS wraps a modified linear Thompson sampling bandit algorithm in a Batch Bayesian optimization method. Essentially, the Batch Bayesian optimization method optimizes the delayed effects correction terms in an outer loop based on returns provided by a Thompson sampling bandit algorithm that leverages the delayed effect correction term. We begin by introducing Thompson Sampling with delayed effect correction. We then present the BOTS algorithm. We conclude this section by presenting the simulation environment we use to evaluate the BOTS algorithm.

### 3.1 Thompson Sampling with Delayed Effect Correction

To model the delayed effect of an action, we introduce a delayed effect parameter $\beta_a$ for each action, which controls how much to modify the the immediate reward by for each action. Within the Thompson sampling algorithm, we replace the computation of the expected reward with the computation of an approximate state-action value function $\boldsymbol{v}_a \leftarrow \hat{\boldsymbol{w}}_a \boldsymbol{x}_s - \beta_a$. The algorithm is sketched in Algorithm 1. The delayed effect correction terms $\beta_a$ are held constant within a given episode of Thompson sampling. However, the proposed BOTS algorithm uses an outer Batch Bayesian optimization loop to learn values for the delayed effect correction terms that maximize the expected return of the Thomson Sampling with delayed effects correction algorithm. We describe the full algorithm in the next section.

### 3.2 The BOTS Algorithm

The BOTS algorithm is detailed in Algorithm 2. We summarize notation in Table A.1. As noted above, at a high level, BOTS wraps the Thompson sampling with delayed effects algorithm with an outer loop of Batch Bayesian optimization to learn the delayed effects correction terms based on Thompson sampling returns. BOTS partitions the total episode budget into an initialization phase followed by a number of rounds $R$ of batch Bayesian optimization where the same batch size $B_R$ is used in each round. On each round, we evaluate $B_R$ different sets of delayed effects correction terms. The performance of a given set of delayed effects correction terms is estimated using the

---

**Algorithm 1** TS with Bayesian linear regression, and delayed effect

**Require:** $n$, priors $\boldsymbol{M}_a$, $\boldsymbol{S}_a$ for $a \in [0, A]$, and $\boldsymbol{r}$ empty array of size $S$
1: $\boldsymbol{x}_s \leftarrow env_n.reset()$
2: $done \leftarrow False$
3: **while** $done$ is not $True$ **do**
4:    **for** $a = 0 : A$ **do**
5:       $\hat{\boldsymbol{w}}_a \sim MVN(\boldsymbol{M}_a, \boldsymbol{S}_a)$
6:       $\boldsymbol{v}_a \leftarrow \hat{\boldsymbol{w}}_a \boldsymbol{x}_s - \beta_a$
7:    **end for**
8:    $a_s^* \leftarrow \arg\max_a \boldsymbol{v}_a$
9:    $\boldsymbol{x}_{s+1}, \boldsymbol{r}_s, done \leftarrow env_n.step(a_s^*)$
10:   $\boldsymbol{M}_{a_s^*}, \boldsymbol{S}_{a_s^*} \leftarrow update\_posterior(a_s^*, \boldsymbol{x}_s, \boldsymbol{r}_s),$
11:   $\boldsymbol{x}_s \leftarrow \boldsymbol{x}_{s+1}$
12: **end while**
13: **return** cumsum($\boldsymbol{r}$), $\boldsymbol{M}_a$, $\boldsymbol{S}_a$ for $a \in [0, A]$

---

**Algorithm 2** BOTS Algorithm: batch BO on delayed effects for TS

**Require:** $R, B_R, B_0, \boldsymbol{M}_0, \boldsymbol{S}_0, l$
1: $N \leftarrow \frac{120}{R \times B_R}$
2: $\mathcal{D}_0, \mathcal{P}_0 \leftarrow initialize(B_0, \boldsymbol{M}_0, \boldsymbol{S}_0)$
3: **for** $r = 1 : R$ **do**
4:    $\tilde{\mathcal{D}} \leftarrow filter(\mathcal{D}_{r-1}, B_R, B_0)$
5:    $GP \leftarrow fit(\tilde{\mathcal{D}})$
6:    $\boldsymbol{\beta} \leftarrow acqfn(GP)$
7:    **for** $b = 1 : B_R$ **do**
8:       $\boldsymbol{M}_{rb}, \boldsymbol{S}_{rb} \leftarrow refine\_prior(\boldsymbol{\beta}_{rb}, \mathcal{P}_{r-1}, l)$
9:    **end for**
10:   **for** $b = 1 : B_R$ **do**
11:      **for** $n = 1 : N$ **do**
12:         $\boldsymbol{Y}_{rbn}, \boldsymbol{M}_{rbn}, \boldsymbol{S}_{rbn} \leftarrow TS(n, \boldsymbol{\beta}_{rb}, \boldsymbol{M}_{rb}, \boldsymbol{S}_{rb})$
13:      **end for**
14:      $\bar{\boldsymbol{Y}}_{rb} \leftarrow \frac{1}{N} \sum_{n=1}^{N} \boldsymbol{Y}_{rbn}$
15:      $\bar{\boldsymbol{M}}_{rb} \leftarrow \frac{1}{N} \sum_{n=1}^{N} \boldsymbol{M}_{rbn}$
16:      $\bar{\boldsymbol{S}}_{rb} \leftarrow \frac{1}{N} \sum_{n=1}^{N} \boldsymbol{S}_{rbn}$
17:   **end for**
18:   $\mathcal{D}_r \leftarrow \mathcal{D}_{r-1} \cup \{(\boldsymbol{\beta}_{rb}, \bar{\boldsymbol{Y}}_{rb}, r) \mid b = 1 : B_R\}$
19:   $\mathcal{P}_r \leftarrow \mathcal{P}_{r-1} \cup \{(\boldsymbol{\beta}_{rb}, \bar{\boldsymbol{M}}_{rb}, \bar{\boldsymbol{S}}_{rb}) | b = 1 : B_R\}$
20: **end for**

---

correspond return from executing the Thompson Sampling with delayed effects correction algorithm in parallel from a specified number of episodes $N$. Here, each episode corresponds to a Thompson Sampling-based adaptive intervention trial applied to a single individual.

BOTS implements this basic idea with two important enhancements. First, the performance of Thompson sampling is heavily dependent on the prior. When using multiple rounds of Batch Bayesian optimization, we have the opportunity to chain the posteriors found in one round to the prior used in the next round. However, this chaining is only reasonable if the values of the delayed effects corrections terms are similar. Therefore, we implement an adaptive prior refinement procedure in algorithm 3. This algorithm identifies the previously evaluated delayed effects terms that are as close as possible to those selected for a new round and copies their corresponding posterior. In the case where $N > 1$, we average the posterior across all the individuals that used the same delayed effects correction terms. We can also filter the candidate previously evaluated delayed effects terms to restrict them to more recent rounds.

However, chaining the Thompson sampler priors across rounds means that the function that the Bayesian optimization method is attempting to approximate will change over time as changing the Thompson sampler prior will change the distribution of returns. To solve this problem, we adopt a basic continual learning modification to the Bayesian optimization process. Specifically, we restrict the update to the GP to use a filtered set of observations from the most recent rounds only. This algorithm in sketched in Algorithm 4.

---

**Algorithm 3** Refine prior for TS coefficients

---

**Require:** $\beta$, $\mathcal{P}_{r-1}$, $l$
1: $k, v \leftarrow \arg\min_{j \in \{1,\dots,B_R\}, s \in \{r-l,\dots,r-1\}} | \beta - \beta_{js} |$
2: **return** $\bar{M}_{kv}, \bar{S}_{kv}$

---

---

**Algorithm 4** Filter: construct training data $\tilde{\mathcal{D}}$ for fitting GP

---

**Require:** $\mathcal{D}_{r-1}, B_R, B_0$
1: **if** $B_R \geq B_0$ **then**
2:     $\tilde{\mathcal{D}} \leftarrow \mathcal{D}_{r-1}$
3: **else**
4:     $K \leftarrow 1$
5:     **while** $KB_R < B_0$ **do**
6:        $K \leftarrow K + 1$
7:     **end while**
8:     $E \leftarrow B_0 - KB_R$
9:     $\mathcal{D}_{full} \leftarrow$ select full batch of data of previous $K$ rounds
10:     $\mathcal{D}_{random} \leftarrow$ select randomly E data from round $K+1$
11:     $\tilde{\mathcal{D}} \leftarrow \mathcal{D}_{full} \cup \mathcal{D}_{random}$
12: **end if**
13: **return** $\tilde{\mathcal{D}}$

---

In this algorithm, $\tilde{\mathcal{D}}$ is the training data used for fitting the GP. We select the most recent data to include in $\tilde{\mathcal{D}}$, such that $\tilde{\mathcal{D}}$ ends up with a size of $\max(B_0, B_R)$. If $B_R$ is smaller than $B_0$, then we first take all the points from the most recent batch, then use data from batches from previous rounds. We make a final random selection among available data from the final round considered such that we have exactly $B_0$ observations in $\tilde{\mathcal{D}}$. For example, to construct $\tilde{\mathcal{D}}$ when $B_R = 4$ and $B_0 = 10$, we first select the four points from the most recent round, then all the four points from the previous round, then 2 points chosen at random from the round before last, thus yielding 10 points in $\tilde{\mathcal{D}}$. If $B_R > 10$, then $\tilde{\mathcal{D}}$ contains all the points from the most recent round.

The initialization phase for BOTS is standard and is sketched in in Algorithm 5. We generates $B_0$ initial training data points in $\mathcal{D}_0$ for initializing the GP.

### 3.3 JITAI Simulation Environment

To evaluate the proposed approach, we use a Just-in-Time Adaptive Intervention (JITAI) simulation environment. The simulator significantly extends that introduced in Karine et al. (2023) by considering the case of stochastic behavioral dynamics as well as distributions over trait-level parameters. The base JITAI environment models a messaging-based physical activity intervention. The state includes a binary context ($C$), habituation level ($H$), disengagement risk level ($D$), and the number of steps ($S$) which is the reward. The true context can be conceptualized as representing a variety of behavioral or activity states such as 'stressed' or 'not stressed'. We summarize the JITAI environment variables in Table A.1.

The simulation includes four possible actions: action 0 indicates that no message is sent to the participant, action 1 indicates that a non-contextualized message is sent to the participant, action 2 indicates that a message customized to context 0 is sent to the participant, and action 3 indicates that a message customized to context 1 is sent to the participant. Thus, the actions are $a \in [0, A]$, where $A = 3$. We summarize the actions in Table A.1. The maximum length is 50 time steps. The simulation runs until it reaches the maximum length, or when the disengagement risk threshold is hit, which ever occurs first.

We create a stochastic version of this environment by introducing noise into the existing deterministic dynamics. Since the habituation and disengagement risk values are in $[0, 1]$, we add noise by sampling from a beta distribution whose expected value is set to the output of the deterministic dynamics. The spread of the distribution is controlled by concentration parameters $\kappa_d$ and $\kappa_h$. We use a similar approach to adding noise to the step count dynamics. However, in this case the step counts are positive only and we model their distribution as a gamma with expected value set to the out out

---

**Algorithm 5** Initialize: generate initial data $\mathcal{D}_0, \mathcal{P}_0$

---

**Require:** input $B_0$, $\boldsymbol{M}_0$, $\boldsymbol{S}_0$
1: **for** $b = 1 : Bo$ **do**
2:     **for** $a = 0 : A$ **do**
3:        $\beta_{ab} \sim Uniform(0, 1)$
4:     **end for**
5:     **for** $n = 1 : N$ **do**
6:        $\boldsymbol{Y}_{bn}$, $\boldsymbol{M}_{bn}$, $\boldsymbol{S}_{bn}$ $\leftarrow TS(n, \beta_b, \boldsymbol{M}_0, \boldsymbol{S}_0)$
7:     **end for**
8:     $\bar{\boldsymbol{Y}}_b \leftarrow \frac{1}{N} \sum_{n=1}^{N} \boldsymbol{Y}_{bn}$
9:     $\bar{\boldsymbol{M}}_b \leftarrow \frac{1}{N} \sum_{n=1}^{N} \boldsymbol{M}_{bn}$
10:     $\bar{\boldsymbol{S}}_b \leftarrow \frac{1}{N} \sum_{n=1}^{N} \boldsymbol{S}_{bn}$
11: **end for**
12: $\mathcal{D}_0 \leftarrow \{(\beta_b, \bar{\boldsymbol{Y}}_b, 0) \mid b = 1 : B_0\}$
13: $\mathcal{P}_0 \leftarrow \{(\beta_b, \bar{\boldsymbol{M}}_b, \bar{\boldsymbol{S}}_b) | b = 1 : B_0\}$
14: **return** $\mathcal{D}_0, \mathcal{P}_0$

---

of the deterministic dynamics and variance equal to $\sigma_s^2$. The distributions are summarized below.

$$\tilde{h}_t \sim Beta\big(\kappa_h h_t, \kappa_h(1 - h_t)\big), \quad \tilde{d}_t \sim Beta\big(\kappa_d d_t, \kappa_d(1 - d_t)\big), \quad \tilde{s}_t \sim Gamma\big(\big(\frac{s_t}{\sigma_s}\big)^2, \frac{\sigma_s^2}{s_t}\big)$$

## 4 EXPERIMENTS

In this section we describe experiments and results. We begin with a description of empirical protocols and algorithm settings used. We then present results. Additional findings are included in the supplemental material.

### 4.1 EMPIRICAL PROTOCOLS

We perform experiments suing the JITAI environment introduced in the previous section. We use the default settings of the base deterministic JITAI environment and the stochastic dynamics parameters $\kappa_h = 50$, $\kappa_d = 50$, $\sigma_s = 25$. We summarize the JITAI environment parameters in Table A.1. The Thompson sampler's initial prior parameters are $\mu_w = 0.$, $\mu_b = 50$, $\sigma_y = 100$, and noise $\sigma_Y = 25$.

In terms of the application of the BOTS algorithm, we study a setting where we set $\beta_0$, the delayed effect associated with not sending a message, to 0. As the remaining actions all have a potential negative delayed effect on future step count, we tie their delayed effects correction terms together yielding $\beta = \beta_1 = \beta_2 = \beta_3$. We experiment with different combinations of $(R, B_R, N)$ configurations such that we conserve the product $N \times R \times B_R = 120$. Recall that $N$ is the number of simulated participants used to evaluate each delayed effect setting, $R$ is the number of BO rounds, and the corresponding $B_R$ is the BO batch size. We run experiments for $N = 1, 5, 10$. For example, for $N = 10$, the possible values for $(R, B_R)$ are: $(1, 12), (2, 6), (3, 4), (4, 3), (6, 2), (12, 1)$.

For the batch Bayesian optimization acquisition function used in BOTS, we choose qEI as implemented in the BoTorch library (Balandat et al., 2020). We performed preliminary experiments using various values of number of previous rounds of observations used when refining the prior Thompson sampler prior. The results for $l = 1$ and $l = 3$ are similar, so we show the results for $l = 1$.

All experiments are repeated five times. We compute the train average return per $(R, B_R)$, for a fixed $N$: $AR_{R,B_R}$, and the test average return per $(R, B_R)$, for a fixed $N_{perf}$: $AR_{R,B_R}$. The test return fixes the optimal delayed effects parameters.

$$AR_{R,B_R} = \frac{1}{T} \sum_{t=1}^{T} \frac{1}{R} \sum_{r=1}^{R} \frac{1}{B_r} \sum_{b=1}^{B_r} \frac{1}{N} \sum_{n=1}^{N} \left( \sum_{s=1}^{S} y_{trbns} \right) \tag{1}$$

$$AR_{R,B_R} = \frac{1}{T} \sum_{t=1}^{T} \frac{1}{N_{perf}} \sum_{n=1}^{N_{perf}} \left( \sum_{s=1}^{S} y_{perftns} \right) \tag{2}$$

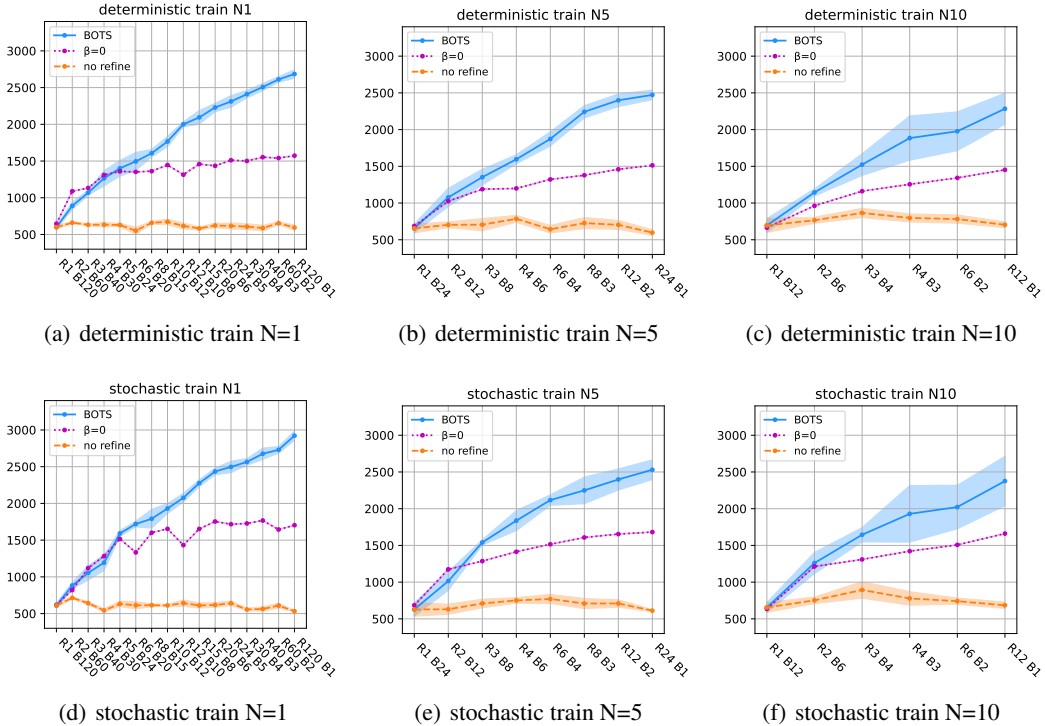

Figure 1: Training average returns for JITAI deterministic env, and JITAI stochastic env, for $N = 1, 5, 10$, using BOTS, $\beta = 0$, and no refine methods.

## 4.2 PERFORMANCE OF BOTS

To begin, we show the base performance of the BOTS algorithm in Figure 1 as the blue lines. The top set of plots evaluate BOTS on the base deterministic JITAI environment, while the bottom set of plots evaluate BOTS on the stochastic environment. The three panels in each row correspond to settings where the returns used to optimize the GP within BOTS is based on an average over $N = 1$, $N = 5$ or $N = 10$ individuals. As noted previously, we hold the total number of simulated individuals constant at 120 and each contributes one episode only. This yields a variety of possible configurations of the number of rounds and the batch size for each value of $N$. We order the x-axis of each plot by the number of rounds.

As we can see in these results, the average training performance increases as the number of rounds increases or equivalently, the batch size decreases. These results suggest that when configuring a real adaptive intervention tuning study, it is beneficial to the participants on average to use a larger number of rounds. This of course trades off with the total available time to conduct a study.

We show the average test performance in Section A.3. Similarly as for the average training performance, the average test performance increases as the number of rounds increases or equivalently, the batch size decreases.

In terms of absolute performance attained, we can compare BOTS to full RL methods. We performed experiments using both classical REINFORCE and deep Q networks as in Karine et al. (2023). The results are shown in Figure 3 in the Appendix. We can see that BOTS can achieve better performance than these methods when we consider total episode count and the ability to perform multiple episodes in parallel.

## 4.3 BOTS ABLATION STUDY

We next repeat the experiments conducted in the previous section using two ablations of the BOTS algorithm. First, we consider the case where we perform multiple rounds of Thompson sampling

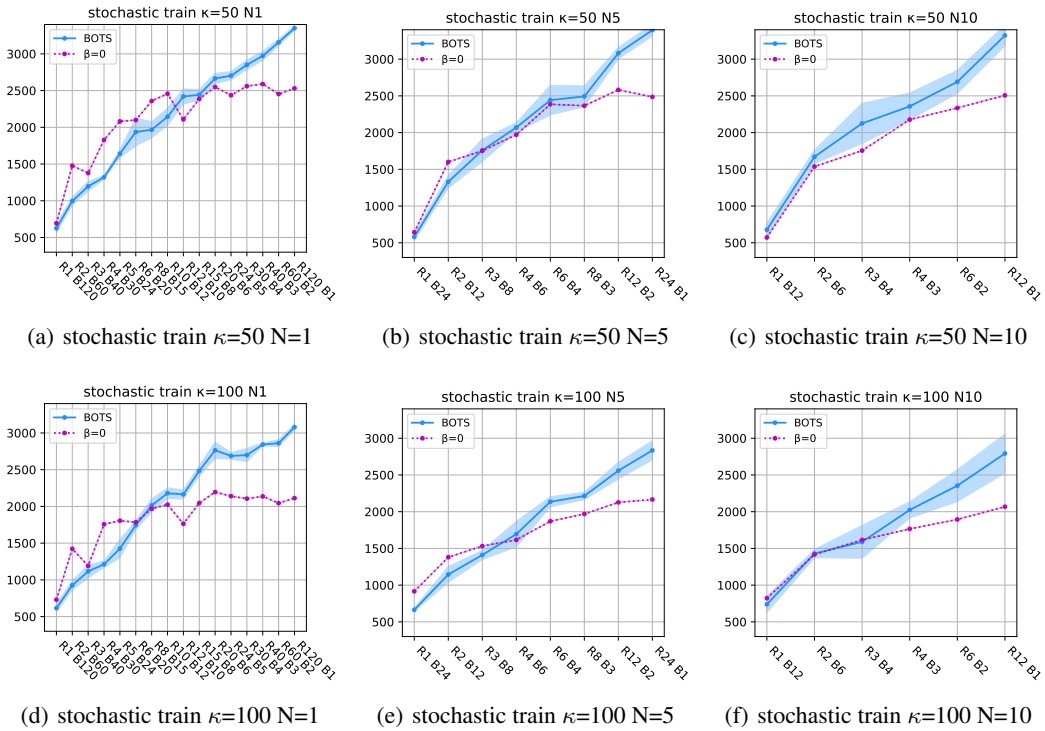

Figure 2: Training average returns for JITAI stochastic env, for $N = 1, 5, 10$, using BOTS and $\beta = 0$, with distributions on $\delta_d \, \epsilon_d$, $\epsilon_h$, and $\delta_h$, for $\kappa = 50$ and $\kappa = 100$.

and chain the posteriors from one round into the prior on the next round. However, we fix the delayed effect parameter to $\beta = 0$. Second, we consider the case where we learn the delayed effects parameter $\beta$ using Bayesian optimization, but do not refine the Thompson sampler prior across rounds. As the in the previous section, we explore different $(R, B_R, N)$ combinations. These results are shown as the magenta ($\beta = 0$) and orange (no prior refinement) lines in figure 1.

As we can see from the results, not refining the prior based on data from prior rounds has a severely negative effect on performance. The magnitude of this effect depends on how informative the initial prior is, and in the setup of these experiments it is set to be very broad. Interestingly, fixing $\beta = 0$ has a less drastic effect on performance, but we can clearly see that when more than a handful of rounds are performed, there is a strong benefit to the optimization of $\beta$ using the full BOTS algorithm.

### 4.4 BOTS AND INCREASING BETWEEN-PERSON VARIABILITY

In the prior experiments, we use the default settings of the JITAI environment dynamics parameters. This includes fixed default values for the hyper-parameters that govern habituation and disengagement increase and decay in response to actions and true contexts. We can evaluate BOTS in a more realistic setting where different individuals undergo not only unique stochastic trajectories in response to time varying contexts and selected actions, but have different dynamics governing habituation and disengagement risk. To this end, we add distributions to the trait-level parameters disengagement risk decay $\delta_d$, disengagement risk increment $\epsilon_d$, habituation decay $\delta_d$, and habituation increment $\epsilon_d$ as follows:

$$\tilde{\delta}_h \sim Beta\big(\kappa_{\delta_h}\delta_h, \kappa_{\delta_h}(1 - \delta_h)\big), \quad \tilde{\epsilon}_h \sim Beta\big(\kappa_{\epsilon_h}\epsilon_h, \kappa_{\epsilon_h}(1 - \epsilon_h)\big)$$

$$\tilde{\delta}_d \sim Beta\big(\kappa_{\delta_d}\delta_d, \kappa_{\delta_d}(1 - \delta_d)\big), \quad \tilde{\epsilon}_d \sim Beta\big(\kappa_{\epsilon_d}\epsilon_d, \kappa_{\epsilon_d}(1 - \epsilon_d)\big)$$

We choose these distributions so that the expected value of the distribution over each parameter will match the base simulator's default values. In Figure 2, we show the results with concentration hyper-parameters set to $\kappa = 50$, and $\kappa = 100$. Interestingly, we can see that the maximum performance

increases somewhat in this setting both for BOTS and for the baseline approach with $\beta = 0$. This is likely due to the fact the the original habituation and disengagement risk dynamics parameters were set quite stringently. A higher return would be expected when any of them are set to less sensitive values. Nonetheless, we can see that BOTS continues to yield strong performance improvements over the $\beta = 0$ case for a range of round and batch sizes.

## 5 CONCLUSIONS

We introduce a novel algorithm BOTS which uses batch Bayesian optimization, and refined priors to automatically capture the delayed effect in Thompson sampling bandits. We show that our algorithm can match the performance of full RL methods using fewer episodes. We further show that both the delayed effect estimation and the prior refinement are key to the success of the algorithm. The results overall suggest that BOTS achieves the best performance in terms of average training return at the maximum number of rounds. However, the decrease in performance appears to be sub-linear with respect to the number of rounds. In real-world settings, this is likely to be a helpful property as running fully sequential optimization will not be realizable given the length of adaptive intervention trials for each individual. In terms of future work, we plan to investigate refinements to the BOTS algorithm that are not restricted to using a constant batch size per round. We also plan to evaluate BOTS on the JITAI environment without the constraint on tied delayed effects terms. This may result in improved performance, but the outcome depends on the resulting bias-variance trade-off as this will increase the dimensionality of the Bayesian optimization search space.

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

# A APPENDIX

## A.1 NOTATIONS

We summarize the variables for BOTS algorithm, the actions, and the variables and parameters for the JITAI environment.

### BOTS algorithm variables

| | |
|---|---|
| $\beta$ | Delayed effect parameter for Thompson Sampling. This controls how much to penalize the reward. |
| $B_0$ | Initial number of random $\beta$ values, in the $intialize(...)$ function. This is the initial number of training data for fitting GP. $B_0 = 10$. |
| $B_R$ | Batch size. $B_R$ varies such that $R \times B_R \times N = 120$. This is the number of candidates chosen by the acquisition function. |
| $d$ | Dimension of the observed data. $d = 3$ since the observed data are $(C, H, D)$. |
| $\mathcal{D}_0$ | Initial training data for fitting the GP. $\mathcal{D}_0 = \{(\beta_b, \bar{Y}_b, 0) \mid b = 1 : B_0\}$. |
| $\mathcal{D}_r$ | Training data for fitting the GP, at round $r$. |
| $l$ | Number of previous rounds that are used to refine the Thompson Sampling priors. This is used in the $refine\_prior(...)$ function. |
| $\boldsymbol{M}_0$ | Initial prior mean for the Thompson Sampling weights and bias. This a matrix of size $\mathbb{R}^{(1+d) \times A}$. |
| $\boldsymbol{M}_{rbn}$ | Posterior mean for the Thompson Sampling weights and bias, at round $r$, batch $b$, environment $n$. This a matrix of size $\mathbb{R}^{(1+d) \times A}$. |
| $\bar{\boldsymbol{M}}_{rb}$ | Average of posterior means for the Thompson Sampling weights and bias, over $N$ environments, at round $r$, batch $b$. This a matrix of size $\mathbb{R}^{(1+d) \times A}$. |
| $\mathcal{P}_0$ | Initial set for refine priors. $\mathcal{P}_0 = \{(\beta_b, \bar{\boldsymbol{M}}_b, \bar{\boldsymbol{S}}_b) \mid b = 1 : B_0\}$. |
| $\mathcal{P}_r$ | Set for refine priors, at round $r$. This is used in the $refine\_prior(...)$ function. This set contains batches of $\beta$'s, and their corresponding average posteriors, at round $r$. |
| $\boldsymbol{S}_0$ | Initial covariance matrix for the Thompson Sampling weights and bias. This a matrix of size $\mathbb{R}^{(1+d) \times (1+d) \times A}$. |
| $\boldsymbol{S}_{rbn}$ | Posterior covariance matrix for the Thompson Sampling weights and bias, at round $r$, batch $b$, environment $n$. . This a matrix of size $\mathbb{R}^{(1+d) \times (1+d) \times A}$. |
| $\bar{\boldsymbol{S}}_{rb}$ | Average of posterior covariance matrices for the Thompson Sampling weights and bias, over $N$ environments, at round $r$, batch $b$. This a matrix of size $\mathbb{R}^{(1+d) \times (1+d) \times A}$. |

**BOTS algorithm variables (continued)**

| | |
|---|---|
| $n$ | Index for the environment $n$ (a.k.a. participant $n$). |
| $N$ | Number of environments (a.k.a participants). $N$ varies such that $R \times B_R \times N = 120$. |
| $R$ | Number of BO rounds. $R$ varies such that $R \times B_R \times N = 120$. |
| $\boldsymbol{Y}_{bn}$ | Initial return at batch $b$, environment $n$. This is obtained by running Thompson Sampling, on environment $n$, inside the $initialize(...)$ function. |
| $\boldsymbol{Y}_{rbn}$ | Return at round $r$, batch $b$, environment $n$. This is obtained by running Thompson Sampling, on environment $n$. |
| $\bar{\boldsymbol{Y}}_{rb}$ | Average return over $N$ environments, at round $r$, batch $b$. |

**Actions**

| | |
|---|---|
| $a$ | Action value, where $a = 0$ indicates that no message is sent to the participant, $a = 1$ indicates that a non-contextualized message is sent to the participant, $a = 2$ indicates that a message customized to context 0 is sent to the participant, and $a = 3$ indicates that a message customized to context 1 is sent to the participant. |
| $A$ | Maximum index for the actions. $A = 3$, and the action $a \in [0, A]$. |

**JITAI environment variables**

| | |
|---|---|
| $C, c_t, \tilde{c}_t$ | $C$ is the true context variable. It has a binary value, thus $c_t = 0$ or 1. $\tilde{c}_t$ is the stochastic version of $c_t$. |
| $H, h_t, \tilde{h}_t$ | $H$ is the habituation level variable. $h_t \in [0, 1]$. $\tilde{h}_t$ is the stochastic version of $h_t$. |
| $D, d_t, \tilde{d}_t$ | $D$ is the disengagement risk variable. $d_t \in [0, 1]$. $\tilde{d}_t$ is the stochastic version of $d_t$. |
| $\kappa_d$ | Disengagement concentration parameter of the $beta$ distribution for $\tilde{d}_t$. |
| $\kappa_h$ | Habituation concentration parameter of the $beta$ distribution for $\tilde{h}_t$. |
| $S, s_t, \tilde{s}_t$ | $S$ is the step count variable. $s_t \in \mathbb{N}$. This is also the reward. $\tilde{s}_t$ is the stochastic version of $s_t$. |
| $\sigma_s$ | Step count noise parameter of the $gamma$ distribution for $\tilde{s}_t$. |

**JITAI environment parameters**

| | |
|---|---|
| $\delta_d, \tilde{\delta}_d$ | Disengagement decay. $\delta_d = 0.1$. $\tilde{\delta}_d$ is a sample from a distribution centered on $\delta_d$. |
| $\delta_h, \tilde{\delta}_h$ | Habituation decay. $\delta_h = 0.1$. $\tilde{\delta}_h$ is a sample from a distribution centered on $\delta_h$. |
| $\epsilon_d, \tilde{\epsilon}_d$ | Disengagement increment. $\epsilon_d = 0.4$. $\tilde{\epsilon}_d$ is a sample from a distribution centered on $\epsilon_d$. |
| $\epsilon_h, \tilde{\epsilon}_h$ | Habituation increment. $\epsilon_h = 0.05$. $\tilde{\epsilon}_h$ is a sample from a distribution centered on $\epsilon_h$. |
| $\kappa_{\delta_d}$ | Disengagement decay concentration parameter of the *beta* distribution for $\tilde{\delta}_d$. |
| $\kappa_{\delta_h}$ | Habituation decay concentration parameter of the *beta* distribution for $\tilde{\delta}_h$. |
| $\kappa_{\epsilon_d}$ | Disengagement increment concentration parameter of the *beta* distribution for $\tilde{\epsilon}_d$. |
| $\kappa_{\epsilon_h}$ | Habituation increment concentration parameter of the *beta* distribution for $\tilde{\epsilon}_h$. |
| $\sigma$ | Feature uncertainty. $\sigma = 0.4$. This is the uncertainty parameter used by the JITAI environment to generate the true context. |

## A.2 LEARNING CURVES WITH NO RESTRICTION ON NUMBER OF ROUNDS

To get some insights on the upper bounds of the returns, we run the RL algorithms: DQN and REINFORCE, on both the JITAI stochastic and deterministic environments. The learning curves show that the training returns converge to $\approx 3000$. The results are shown in Figure 3.

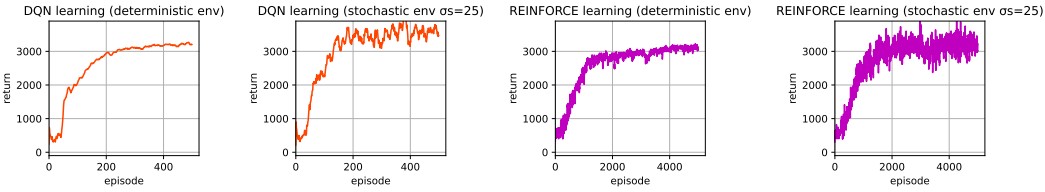

Figure 3: Learning curves when using RL, with JITAI deterministic env and stochastic env.

## A.3 PERFORMANCE RETURNS PLOTS

We show the plots for the test average returns. The average test performance increases as the number of rounds increases or equivalently, the batch size decreases.

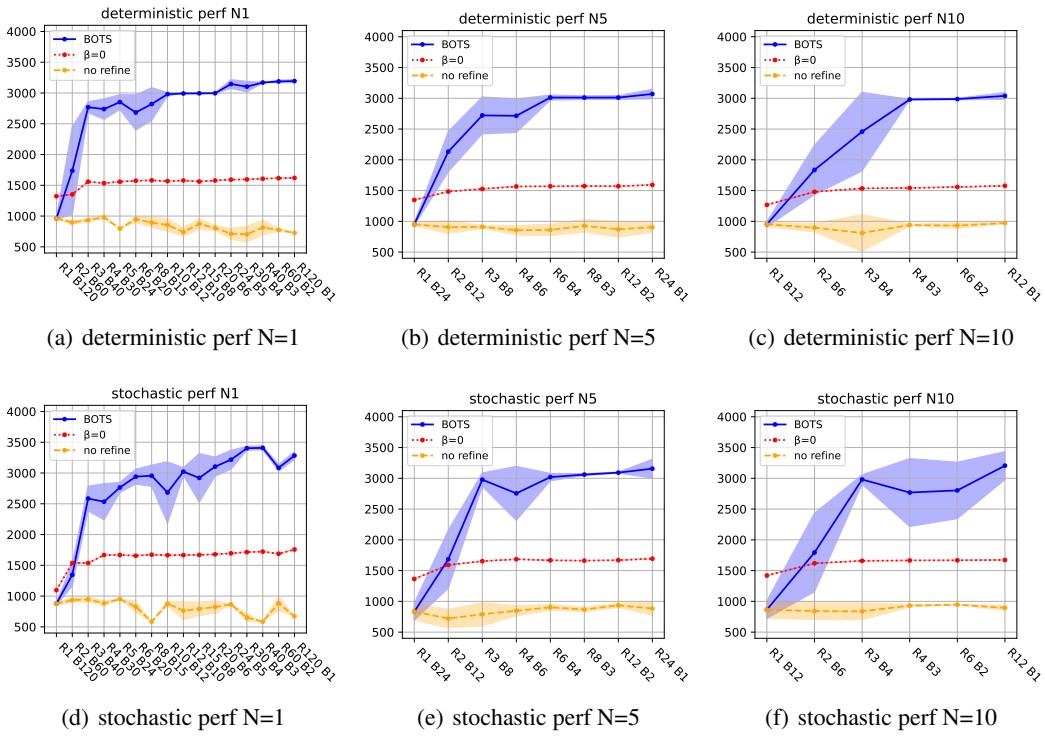

Figure 4: Test average returns for the JITAI deterministic env, and JITAI stochastic env, for $N = 1, 5, 10$, using BOTS, $\beta = 0$, and no refine methods.

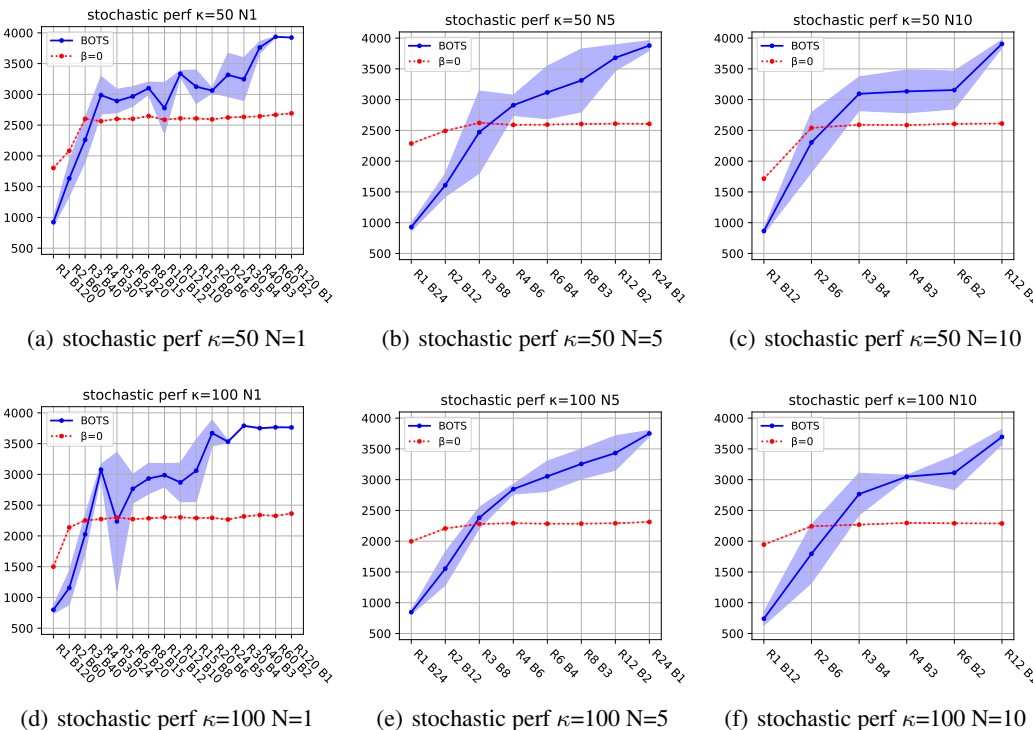

Figure 5: Test average returns for the JITAI stochastic env, for $N = 1, 5, 10$, using BOTS and $\beta = 0$, with distributions on $\delta_d$ $\epsilon_d$, $\epsilon_h$, and $\delta_h$, for $\kappa = 50$ and $\kappa = 100$.

