# OpenReview forum: "Batch Bayesian Optimization of Delayed Effects Corrections for Thompson Sampling Bandits: A Practical Tuning Algorithm for Adaptive Interventions"
_ICLR.cc/2024/Conference — ICLR 2024 Conference Withdrawn Submission_

### Official Review · Reviewer_UpDA · 2023-10-24

**Soundness:** 1 poor
**Presentation:** 2 fair
**Contribution:** 2 fair
**Rating:** 3
**Confidence:** 4

**Summary:**

The Thompson sampling (TS) method has been used in the healthcare community for achieving better bias-variance trade-off compared to conventional RL methods. However, in the healthcare area, each TS round usually involves one or more humans such that the trial is costly and incurs delayed effects of actions. To resolve this issue, this work proposed to capture the delayed effects of each action using a vector of delayed effect correction parameters and optimize them using batch Bayesian optimization (BO). Some tricks such as warm-starting are applied to TS for improving the performance. The proposed method is tested on a JITAI simulation environment.

**Strengths:**

This paper introduced an important and interesting problem when applying bandit algorithms to the healthcare community. It contributes a new idea on modeling and capturing the delayed effects of human actions in the bandit process, which may help to improve the usability of bandit methods in real-world healthcare applications.

**Weaknesses:**

1. The novelty of the proposed method is limited. The proposed method is a straightforward application of conventional batch BO to optimize the delayed effects correction terms in TS. I didn't identify any novel technical issues or challenges for doing this. Also, the motivation for choosing batch BO is not clear. Since the relationship between the $v_a$ and $\beta_a$ has been formulated (line 6 in Algorithm 1), why is a black-box optimization algorithm (i.e., BO) selected? Why not learn $\beta_a$ directly using Bayesian inference, SGD, or any other optimization method?

 2. I have serious concerns about the soundness of the experiments:

- Firstly, the proposed method is not fairly compared with the baselines. Even though the results of DQN and REINFORCE are shown in Appendix A.2, the figures only include the learning curve of the baseline and don't show that of the proposed method. In the current presentation, I don't know how to achieve the conclusion that "BOTS can achieve better
performance than these methods ...". For example, in the first two graphs of Fig. 3, it seems that DQN can achieve a return larger than 2000 with only 100 episodes, which is better than many blue points in Fig. 1. I suggest the authors to re-organize the graphs such that the comparisons between different results can be straightforward.

- Secondly, one key contribution of the proposed method is to optimize the delayed effects correction terms using BO. To demonstrate the priority of the proposed method, some simple baselines (such as the conventional TS, TS+Bayesian inference, TS+random search, etc.) can be considered. Without comparing with suitable baselines, it's hard to measure the significance of the proposed method.

- Besides, from the results shown in Figs. 1&2, it seems that batch BO doesn't work in the proposed method. In all the experiments, the reward increases almost linearly in the number of BO rounds, which are negative observations for supporting the usage of batch BO in this scenario and contradict to the motivation (i.e., execute TS trials in parallel) of this work.

3. The writing of this paper needs to be improved. There are many undefined, inconsistent notations, and typos that make it hard to follow. For example, $x_s$ in Algorithm 1, $\sigma_y\ $, and $\sigma_Y$ in Section 4.1 are not defined. Do $x_s$ and $s_t$ denote the same variable? The notation r is used to represent both the reward and the round index. Some examples of grammatical errors and typos are: "... approach addresses the problem warm-starting the ...", "by for each action", "correspond return", "suing", "As the in the ...", etc.

**Questions:**

1. At the end of Section 2.1, this paper mentioned that existing work requires a number of assumptions about the delayed effect process while this work doesn't rely on any assumption. However, representing the delayed effect using a constant term seems to be a strong assumption on the delayed effect process. Can you provide a detailed analysis of the implicit/explicit assumption used in this work?

2. Why did you assume "$\beta_1 = \beta_2 = \beta_3$" in the experiments? What's the problem of optimizing them separately?

---

### Official Review · Reviewer_ZL6J · 2023-10-28

**Soundness:** 2 fair
**Presentation:** 2 fair
**Contribution:** 2 fair
**Rating:** 3
**Confidence:** 3

**Summary:**

The goal of the study is to learn a decision-making policy for healthcare intervention that can account for delayed feedback, which is the norm in real-life settings.
Existing tools do not work out of the box on this problem: reinforcement learning methods take too many training episodes to learn a good policy, but generating episodes in this setting is very costly, as it involves human subjects; traditional policies from the bandit literature do not account for delayed feedback.
This paper addresses this gap by proposing a novel policy-learning workflow: they first parameterize the delayed effect of an action with a linear model and seek to learn the parameters of this model using Bayesian optimization (BayesOpt) by designing adaptive experiments; the learned model is then used by the bandit Thompson sampling (TS) policy to generate the final decisions to be made.
The final algorithm is shown to work better than non-adaptive variants of the algorithm that either do not account for the delay in observations or do not refine the model used by TS.

**Strengths:**

The paper studies an important problem with realistic settings.
The experiment section is quite thorough, including a wide range of evaluation settings, and the result is conclusive in showing that the proposed policy works better than sample-inefficient RL methods.

**Weaknesses:**

I find that the paper did not do a good job motivating the solutions presented.
Delayed feedback in experimental design has been extensively studied in previous work, but it seems the authors immediately settled on penalizing penalizing immediate reward to account for the delayed effects of an action (which I'm personally still not sure how that works).
I believe a more thorough exposition of this idea and why it is more promising that other ways of accounting for delayed feedback would benefit the paper.

One of the main contributions of the paper is the adaptive refinement of the prior used by the bandit TS policy, but I couldn't find any relevant discussion on this prior.
My understanding is that TS samples from the Bayesian linear regression model whose parameters we are learning each round.
It is not clear what role this extra prior plays.

**Questions:**

- Is the linear assumption realistic?
In which situations this assumption is typically violated?
- In Section 3.2, the authors mention that refining the prior for TS will change the objective function that is being optimized by the inner BayesOpt routine from iteration to iteration.
How does restricting the training data to those from the most recent iterations help with this exactly?
- The experiment section compares the algorithm against a variant that sets $\beta = 0$.
Can we also compare against a range of values for $\beta$?

---

### Official Review · Reviewer_8NfA · 2023-10-29

**Soundness:** 2 fair
**Presentation:** 1 poor
**Contribution:** 1 poor
**Rating:** 1
**Confidence:** 4

**Summary:**

The authors propose a novel menthod to deal with delayed rewards in a bayesian optimization setting. The propose algorithm is a wrapper over the classical linear TS one and works in round, having a specific mechanism to deal with delay.

**Strengths:**

The paper presents a novel algorithm to deal with an interesting problem.

**Weaknesses:**

I think that the paper is not yet ready for publication. There are three main issues:
- there is no theoretical guarantee for the algorithm, therefore so far can be considered only as a heuristic
- the experimental results are not convincing, lacking a proper and fair comparison with the stat of the art
- the paper requires major rewriting since it lacks many details (both in the methodological and in the experimental part.)

Details:
The paper lacks a proper formulation of the BO problem.
In the background section, you mentioned some methods but you did not provide references for them.
What do you mean with for a \in [0, A]
It is hard to get all the details from the description you provide of the algorithm. You should be more specific in the description and add an explanation of the pseudocode line-by-line. Moreover, some of the quantities that are present in the different algorithms are not defined, nor they are present in the final notation table.

The experimental part of the paper lacks a proper baseline comparison. I think that comparing your algorithm only with other versions of your algorithm is weak proof that what you proposed is working.
Some comparison has been proposed in the appendix, but they should be included in the main paper.
Moreover, the comments on the results are not detailed at all. I suggest to try to go deeper and provide more comments on the results you obtained.
You mention that your experiments were produced by repeating the run 5 times and you show some uncertainty regions in the figure. What did you provide there? Standard deviations? Confidence intervals?

**Questions:**

Please refer to the weaknesses.